# Proteobacteria Overgrowth and Butyrate-Producing Taxa Depletion in the Gut Microbiota of Glycogen Storage Disease Type 1 Patients

**DOI:** 10.3390/metabo10040133

**Published:** 2020-03-30

**Authors:** Camilla Ceccarani, Giulia Bassanini, Chiara Montanari, Maria Cristina Casiraghi, Emerenziana Ottaviano, Giulia Morace, Giacomo Biasucci, Sabrina Paci, Elisa Borghi, Elvira Verduci

**Affiliations:** 1Department of Health Sciences, Università degli Studi di Milano, 20142 Milan, Italy; camilla.ceccarani@unimi.it (C.C.); giulia.bassanini@unimi.it (G.B.); emerenziana.ottaviano@unimi.it (E.O.); giulia.morace@unimi.it (G.M.); elvira.verduci@unimi.it (E.V.); 2Institute of Biomedical Technologies, National Research Council, 20090 Segrate, Italy; 3Department of Pediatrics, San Paolo Hospital, 20142 Milan, Italy; cmontanari2@gmail.com (C.M.); sabrina.paci@asst-santipaolocarlo.it (S.P.); 4Department of Food, Environmental and Nutritional Sciences, Università degli Studi di Milano, 20133 Milan, Italy; maria.casiraghi@unimi.it; 5Department of Pediatrics and Neonatology, Guglielmo da Saliceto Hospital, 29121 Piacenza, Italy; g.biasucci@ausl.pc.it; 6Department of Pediatrics, Vittore Buzzi Children’s’ Hospital-University of Milan, 20154 Milan, Italy

**Keywords:** glycogen storage disease, GSD, diet, gut microbiota, short-chain fatty acids, inflammation

## Abstract

A life-long dietary intervention can affect the substrates’ availability for gut fermentation in metabolic diseases such as the glycogen-storage diseases (GSD). Besides drug consumption, the main treatment of types GSD-Ia and Ib to prevent metabolic complications is a specific diet with definite nutrient intakes. In order to evaluate how deeply this dietary treatment affects gut bacteria, we compared the gut microbiota of nine GSD-I subjects and 12 healthy controls (HC) through 16S rRNA gene sequencing; we assessed their dietary intake and nutrients, their microbial short chain fatty acids (SCFAs) via gas chromatography and their hematic values. Both alpha-diversity and phylogenetic analysis revealed a significant biodiversity reduction in the GSD group compared to the HC group, and highlighted profound differences of their gut microbiota. GSD subjects were characterized by an increase in the relative abundance of *Enterobacteriaceae* and *Veillonellaceae* families, while the beneficial genera *Faecalibacterium* and *Oscillospira* were significantly reduced. SCFA quantification revealed a significant increase of fecal acetate and propionate in GSD subjects, but with a beneficial role probably reduced due to unbalanced bacterial interactions; nutritional values correlated to bacterial genera were significantly different between experimental groups, with nearly opposite cohort trends.

## 1. Introduction

The evidence of interplay between intestinal commensal bacteria and host physiological functions has hugely grown over the last years, shedding new light on clinical research on pathological conditions [1]. Among them, inherited metabolic disorders have been shown to be related to gut microbiota composition [2], possibly for the crucial role that diet plays both in patient treatment and in microbial metabolite production. Here, we present our study involving patients affected by glycogen storage diseases (GSD) following the vitally specific diet.

GSD are a group of hereditary metabolic disorders caused by the deficiency of one of the enzymes involved in glycogen metabolism. Glycogen is primarily stored in liver and muscle, and disorders of glycogen degradation may affect both tissues [3,4]. GSD types, grouped by the enzyme deficiency, were numbered as they were discovered, classifying them from GSD type I (von Gierke disease) to GSD type XI [5,6]. The present study focused our research on GSD-I, one of the most common types of glycogen storage diseases.

GSD-I results in a defect in the glucose-6-phosphatase system, which is required for the hydrolysis of glucose-6-phosphate into glucose and inorganic phosphate [7,8], impairing free glucose availability during fasting and glucose homeostasis with consequent hypoglycemia. The clinical onset of GSD-I usually occurs in the first year of life, during complementary feeding, with symptoms related to severe fasting hypoglycemia, hepatomegaly, failure to thrive and growth retardation. The overall annual incidence is about 1 to 100,000 subjects [9,10].

Two main subtypes of GSD-I are recognized: type Ia (GSD-Ia), due to a defect of the catalytic subunit glucose-6-phosphatase-α in the endoplasmic reticulum, and responsible for 80% of cases of GSD-I [3], and type Ib (GSD-Ib), due to a defect of the glucose-6-phosphate translocase, the transporter for the entrance of glucose-6-phosphate into the endoplasmic reticulum [9]. Patients with GSD-Ib may be clinically and metabolically identical to those with GSD-Ia (showing typical physical findings, including protuberant abdomen, truncal obesity, doll-like faces, short stature and hypotrophic muscles [6]), but in addition, most patients with GSD-Ib develop neutropenia and neutrophil dysfunction that predispose them to severe infections and to inflammatory bowel disease (IBD) [11,12]. Although the development of IBD is associated to GSD-Ib, few cases of IBD were recently reported in GSD-Ia [13,14]. Dietary treatment is the cornerstone of GSD-I therapy, and it starts at diagnosis and is life-long. This regimen is characterized by small frequent meals high in complex carbohydrates (preferably with high fiber content) distributed over 24 h [15], including the night, and/or continuous feeding through nasogastric tube [16]. Thus, over the total amount of daily energy intake, the carbohydrate consumption is 60–70%, while 10–15% of calories are derived from proteins and the remaining calories from fat [17,18]. Raw cornstarch is typically introduced between 6 months and 1 year of age [15], since its slow digestion can provide a steady intestinal release of glucose, maintaining more stable glucose levels over a longer period of time [19]. The restriction in sugar consumption is also crucial in the GSD-I diet, since fructose and galactose are metabolized to glucose-6-phosphate and can further contribute to the abnormal biochemical profile; in particular, to hyperlactacidemia [17,19].

The primary aim of the dietary treatment is not only avoiding hypoglycemia, but also achieving a good metabolic control [20], minimizing the secondary metabolic derangements and reducing long-term complications. In order to prevent or treat some clinical conditions (proteinuria, osteoporosis) or biochemical abnormalities (hyperuricemia, hyperlipidemia), patients also take medications/supplementations such as an ACE inhibitor, allopurinol, fibrate, oil fish, calcium and vitamin D3 [16]. GSD-Ib patients also assume granulocyte colony-stimulating factor (G-CSF) and anti-inflammatory drugs to treat neutropenia and IBD, respectively.

Since nutritional intake is one of the most relevant factors influencing the gut microbiota’s composition [21], it is reasonable to expect that such a peculiar diet, along with the daily supplementations, could impact substrates’ availability for microbial fermentation, affecting the production of metabolites; in particular, short chain fatty acids (SCFAs). SCFAs, mainly represented by acetate, propionate and butyrate, are the end products of microbial fermentation in the gastrointestinal tract [22]. Their production is heavily influenced by bacterial cross-feeding interactions, in which acetate and other small molecules (i.e., lactate and succinate) act as substrates to produce butyrate and propionate, respectively [23]. SCFAs are suggested to be involved in the maintenance of the gut barrier function and in the promotion of gut homeostasis [24]. To date, there is no information about gut bacterial metabolite production and consumption regarding GSD microbiota.

The aim of our study was thus to compare dietary macronutrient intake, gut microbial biodiversity and microbial metabolite production in patients with GSD-Ia/Ib and healthy subjects, in order to better evaluate and characterize diet or disease-related microbiome differences.

## 2. Results

### 2.1. Cohort Description

Overall mean BMI values for the enrolled subjects were 26.8 ± 4.8 for GSD patients and 21.6 ± 2.9 for healthy controls (HC) (*p* = 0.0176). Within the entire dataset, 3/21 resulted obese (3/9 GSD, 0/12 HC), 4/21 overweight (3/9 GSD, 1 of which <18 years; 1/12 HC), 14/21 normal weight (3/9 GSD, 11/12 HC). 

All GSD patients were taking drugs to prevent disease-related comorbidities. The reported medications/supplementations were: allopurinol (Ia = 3/4; Ib= 5/5), antihypertensive drugs (Ia = 1/4; Ib = 4/5), triglyceride lower-drugs (Ia = 1/4; Ib = 2/5), salicylates (Ia = 0/4; Ib = 2/5), granulocyte-colony stimulating factor (Ia = 0/4; Ib = 3/5) and multivitamin and calcium with vitamin D (Ia = 4/4; Ib = 5/5).

Three GSD-Ib patients were reported to be neutropenic and to have IBD.

Fasting blood samples of GSD patients were analyzed for total cholesterol, triglycerides, insulin, glucose, uric acid, liver enzymes and lactate (Appendix A). GSD patients showed slightly increased alanine aminotransferase (ALT, mean ± SD: 54.1 ± 43.44 U/L) and aspartate aminotransferase (AST, 42.5 ± 23.8 U/L) values compared to physiological levels (0–35 U/L). In particular, GSD-Ia showed higher values in both parameters (54.5 ± 28.3 U/L and 67.7 ± 47.1 U/L, respectively). GSD-Ia patients showed higher values of both total cholesterol and triglycerides (265.5 ± 152.2 mg/dL) and 422.5 ± 241.6 mg/dL, respectively), compared to normal levels (< 200 mg/dl and < 150 mg/dl, respectively). Serum lactate was also increased in GSD-Ia patients (3.8 ± 1.9 mmol/L) compared to normal levels (0.7–1.15 mmol/L).

### 2.2. Dietary Assessment

The daily energy intakes and the diet macronutrient compositions of enrolled subjects are reported in Table 1.

Compared to HC, GSD group showed a significantly higher daily energy (*p* = 0.0468) and carbohydrate intakes (both grams and % total energy, *p* = 0.002), but a lower lipid intake (% of total energy, *p* = 0.0013) was observed. No significant differences were observed for proteins. As expected from the dietary recommendations, sugar consumption was reduced in the GSD group (*p* = 0.0013), whereas the starch intake was higher in GSD (mean ± SD: 110.27 g ± 44.80) compared to HC (180.94 g ± 62.81) (*p* = 0.004). Total fiber intake (*p* = 0.0148) and soluble fiber intake (*p* = 0.0227) were higher in GSD patients, whereas no significant differences were detected for the insoluble fraction.

### 2.3. Microbiota Profiling

To avoid biases related to uneven sequencing depth (raw reads ranging from 56,150 reads to 350,680), samples were subsampled to 50,000 reads each by random picking. After quality filtering processes, we obtained a mean count of 40,988.261 reads per sample (total count of Operational Taxonomic Units (OTUs) for the entire dataset, average 1654 OTUs per sample).

As shown in Figure 1A, alpha-diversity showed a significant lower biodiversity within GSD subjects for each metric used (chao1, *p* = 0.02; observed species, *p* = 0.02; Shannon, *p* = 0.002; Faith’s phylogenetic diversity, *p* = 0.03). 

A clear difference among HC and GSD subjects was highlighted in beta-diversity as well (Figure 1B). Both unweighted and weighted Unifrac distances revealed a significant separation between groups (respectively, *p* = 0.004 and *p* = 0.01). 

#### 2.3.1. Taxonomic Characterization

We found several significant differences in taxas’ relative abundances among the two groups across all phylogenetic levels. 

At the phylum level (Figure 2A), differences were found in the relative abundance of Firmicutes (GSD 55.9% vs. HC 70%, although not significant) and Proteobacteria (GSD 17% vs. HC 1.4%, *p* = 0.001). Several dominant families were also significantly diverse in the two cohorts: *Ruminococcaceae* (*p* = 0.002), *Veillonellaceae* (*p* = 0.030) and *Enterobacteriaceae* (*p* = 0.006) (Figure 2B). Note, while *Ruminococcaceae* was more abundant among controls (40.3% vs. 15% GSD), both *Veillonellaceae* and *Enterobacteriaceae* were much higher among GSD patients (respectively, 13.8% and 16.3% compared to 4.7% and 1.1% among HC). As shown in Figure 2C, at the genus level GSD patients were severely and significantly depleted in *Ruminococcus* (1.4% vs. 7.2% in HC; *p* = 0.0173), *Faecalibacterium* (7.4% vs. 19.6%; *p* = 0.0209) and *Oscillospira* (0.6% vs. 3.3%; *p* = 0.0020). In total, 1596 OTUs out of were classified as *Enterobacteriaceae* at the family level, and 792 of them were annotated as *Escherichia coli*. GSD patients were found significantly increased in *Escherichia coli* compared to HC (10% vs. 0.93%, *p* = 0.0077).

All relative abundances and significant *p* values are reported in Table 2. 

#### 2.3.2. Fecal Microbial Metabolites

Gas-chromatography analysis revealed an increased production of total fecal short chain fatty acid (SCFA) in GSD group (*p* = 0.0159). In particular, the concentration of acetate and propionate were higher in patients (*p* = 0.031 and *p* = 0.038, respectively), whereas the concentration of butyrate was similar in the two groups (*p* = 0.8381). No significant differences were found for the branched-chain fatty acids iso-valerate and iso-butyrate between the two groups (Appendix A).

#### 2.3.3. Functional Prediction

At a broad functional level (level 2 KEGG), the functional analysis predicted an enrichment in genes encoding enzymes for amino acid metabolism (*p* = 0.0094); in particular, tryptophan (*p* = 0.017), glutathione (*p* = 0.009) and beta-alanine (*p* = 0.0004); and for lipid metabolism, alpha-linolenic acid especially (*p* = 0.025). Intriguingly and counterintuitively, the starch and sucrose metabolism pathways were significantly reduced in GSD subjects (respectively, *p* = 0.026). 

### 2.4. Relationship between Microbial Population, Metabolite Content and Diet

#### 2.4.1. Gut Microbiota and Fecal Microbial Metabolites

A correlation analysis was applied to investigate possible associations between SCFA concentration and bacterial taxa, as shown in Figure 3. 

The obtained data revealed positive correlations between the *Blautia*, *Dorea* and *Phascolarctobacterium* genera, increased in GSD patients, and propionate concentration (R = 0.61, R = 0.82, R = 0.71, respectively). On the other side, we observed *Faecalibacterium* and *Oscillospira* (significantly decreased in GSD subjects) to be negatively related to acetate concentrations (R = −0.47, R = −0.51). 

#### 2.4.2. Impact of Diet on Microbial Taxa Relative Abundance

Correlations of nutritional parameters to bacteria abundances revealed several divergent relationships between the two cohorts, as shown in the Figure 4. 

Fiber intake showed a strong positive correlation to *Odoribacter* and *Parabacteroides* genera (total fibers: R = 0.79, R = 0.78; insoluble fibers: R = 0.79, R = 0.81) only in GSD patients, whereas a milder, opposite trend characterized the HC group. In GSD patients, starch intake positively correlated with *Veillonella*, *Citrobacter* and *Akkermansia* genera (R = 0.299, R = 0.334 and R = 0.406, respectively) and negatively with *Coprococcus* and *Clostridium* genera (R = −0.826 and R = −0.823, respectively). The latter two genera, in particular, showed an opposite trend with nutritional values between the two groups (correlated positively to HC and negatively to GSD patients).

## 3. Discussion

Our study investigated the impact of the life-long cornstarch-rich diet characterizing the treatment of GSD patients by integrating gut microbiota, microbial metabolites and nutritional data. The identification of bacterial metabolism is crucial for the understanding of a possible microbial role in metabolic diseases. For this reason, we highlighted the importance of short chain fatty acids in gut microbiota characterization as bacteria cooperate and feed one on another’s products (cross-feeding). To our knowledge, this is the first study evaluating the impact of the cornerstone diet of GSD-I on gut microbial cross-feeding and metabolites production. 

During the last decade, a healthy gut microbiota has been typically characterized by members of the phyla Bacteroidetes and Firmicutes, and their genera are believed to be the main responsible bacteria for positive biodiversity in the human gut [25], as their balanced abundances and metabolite productions protect the intestinal trait, help digestion and modulate the host innate immune system [26]. In agreement with Colonetti and coworkers [27] that have analyzed the gut microbiota of different types of GSDs, we found a strong reduction in intestinal microbiota richness and diversity compared with healthy controls and a dramatic increase in the phylum Proteobacteria.

Though the GSD diet is enriched in starch and fibers, usually considered good substrates promoting beneficial microbes’ growth, Proteobacteria, in particular, the *Enterobacteriaceae* family, have been suggested to exert pro-inflammatory activity both locally, at the gastrointestinal mucosa level, and systemically [28]. In turn, an inflamed gut seems to constitute a commending environment for proliferation of *Enterobacteriaceae* bacteria [29], and it is also characterized by a depletion of obligate anaerobes, typically recognized as fiber-degrading bacteria [30]. Although GSD-Ib are genetically at risk of intestinal bowel inflammation [16] and three GSD-Ib patients in our dataset were indeed affected, the enrichment in the relative abundance of *Escherichia coli* spanned both type Ia and Ib patients. This data could account for the increased abundance in genes for glutathione metabolism in GSD, since *E. coli* accumulates the tripeptide in order to protect itself from chemical and environmental stress [31]. Despite the high amount of starch in GSD diet, we predicted a reduction of the starch and sucrose metabolism genes, which could be linked to a possible intestinal imbalance caused by both the Proteobacteria’s abnormal abundance and the decrease of obligate anaerobes. Within an inflamed environment, the availability of simple sugars could be altered, and bacteria forced to exploit other nutritional sources such as amino acids. *Enterobacteriaceae*, as seen for the strain *E. coli* LF82, associated to Crohn’s disease, seem to be able to catabolize dietary L-serine in order to maximize their growth [32]. On the other hand, GSD group showed increased amino-acid metabolism genes compared to HC, suggesting that *Enterobacteriaceae* may contribute to the increment of this pathway. Other taxonomic indicators of inflammatory status in GSD patients were the enrichment of the *Blautia* genus, known to stimulate cytokines secretion by host cells [33], and the significant depletion of *Oscillospira* and *Faecalibacterium* species. Data about the observed relative abundance of *Faecalibacterium* spp. and *Escherichia* spp. are in agreement to what Grabherr and colleagues [34] have observed in non-alcoholic steatohepatitis (NASH). Of note, both GSD and NASH are affected by liver damage, and the elevated ALT values found in our patients’ blood samples confirms this similarity.

The depletion in *Faecalibacterium* and *Oscillospira* spp. is a hallmark for patients gut microbiota alteration, as these genera are considered as biomarkers of intestinal and host wellness. *Faecalibacterium* spp. has the ability to produce anti-inflammatory molecules [35], and also a specific protein able to block NF-κB pathway [36]. The genus *Oscillospira* has been found to be constantly reduced in inflammatory diseases as well as *Faecalibacterium* spp. with decreased abundances in Crohn’s disease, both colonic and ileal [37], and in pediatric nonalcoholic steatohepatitis [38]. Moreover, several studies associated *Oscillospira* spp. to lower BMI and leanness-promoting bacteria such as *Christensenella minuta* [39,40]. Our data confirmed these observations, since we observed a higher prevalence of obesity/overweight in GSD cohort compared to HC, and a depletion of *Christensenella* spp. in the patients’ gut. Colonetti and coworkers [27] did not observe changes in *Faecalibacterium* and *Oscillospira* relative abundances in their dataset, and found *Blautia* spp., enriched in our cohort, to be depleted in GSD patients. These differences could be ascribable to the multiplicity of GSDs (type Ia/Ib, III, IX vs. type Ia/Ib), the use of antibiotics before sampling (10/24 subjects vs. 0/9), the sequencing method used (Ion Torrent vs. Illumina MiSeq) and the database used for OTU processing (SILVA vs. Greengenes).

Byndloss and coworkers demonstrated, in vitro and in vivo, the existence of a vicious cycle encompassing the depletion of butyrate-producing microbes and the increase of *Enterobacteriaceae* in the gut microbiota [41]. Indeed, antibiotic-driven reduction of *Lachnospiraceae* and *Ruminococcaceae*, major butyrate producers, promotes the use of glucose instead of butyrate by colonocyte. This metabolic switch-anaerobic glycolysis-fails to suppress host-derived nitrate and oxygen production, promoting the growth of facultative anaerobes such as *Enterobacteriaceae*. On the other hand, the decrease of butyrate downregulates Tregs and epithelial PPAR-γsignaling further promotes the epithelial dysfunction.

We conducted this research considering that the bioavailability of substrates introduced with the diet drives the gut’s microbial composition, and consequently, alterations in intestinal microbiota can lead to a different production of microbial metabolites. 

For instance, the important role *Faecalibacterium* spp. plays in gut microbiota is directly linked to the production of butyrate, the main energy source for enterocytes with a protective role in colorectal cancer and in IBD [42]. On the other hand, the decrease of these bacteria in GSD gut microbiota did not result in a reduction of fecal butyrate concentrations, found in similar amounts in GSD and controls, whereas it could have caused the higher acetate quantities, since this fatty acid has been less used in fermentation reactions. The negative correlation we found between *Faecalibacterium* spp. and acetate concentrations leads in that direction. 

Compared to HC, GSD patients were observed to have a higher concentration of total SCFAs; in particular, acetate and propionate. Those SCFAs have key-roles in gut microbial composition: for instance, acetate production is strongly regulated by the cross-feeding within the gut microbial community. Indeed, Samuel and Gordon [43] underlined that the co-colonization with *Bacteroides thetaiotaomicron*/*Methanobrevibacter smithii* increased serum acetate levels compared to *B. thetaiotaomicron* alone in gnotobiotic mice.

The higher acetate concentration can be ascribed to several intestinal bacteria found to be more abundant in GSD gut microbiota, including *Akkermansia muciniphila* and *Bifidobacterium* spp., which produce acetate by fermenting acetogenic fibers, and to a lesser extent, protein-derived peptides [44].

As well as acetate, propionate seems to exert protective and anti-inflammatory activities in IBD, ameliorating the intestinal mucosa lesion [45]. The higher propionate concentration is in agreement with the enrichment in *Veillonellaceae* family among GSD patients [46]; the genus *Veillonella* was found positively related to starch intake in patients, probably because of their peculiar diet. GSD patients were also enriched in *Megasphaera* genus, which is able to produce propionate from lactate through acrylate pathway [46].

Considering the opposite trends observed between GSD patients and HC, our results indicate that GSD patients have an ongoing alteration in gut microbial community cross-fed by increased pro-inflammatory genera and decreased beneficial bacteria. The specific dietary treatment does not seem to help the composition of gut microbial community in patients, as the anti-inflammatory genera were depleted and not sufficient to counterbalance the dysbiosis. Probiotics supplementation could offer another way to improve and ameliorate the gut microbial population in GSD patients. Carnero-Gregorio and colleagues [14] recently reported a prospective case study pointing toward this direction: by testing a probiotic mixture in a patient with GSD-Ia and Crohn-like IBD, the authors observed a reduced number of bowel episodes and an improvement the patient’s quality of life. Moreover, they found a reduction in *Enterobacteriaceae* relative abundance after the probiotic treatment.

Since patients are commonly taking multiple drugs to cope with the variety of comorbidities characterizing glycogen storage diseases, we tried to evaluate their possible impacts on the microbial community. Indeed, multi-drug usage has been reported to impact microbial composition and richness, but it is difficult to assess whether the observed alterations are caused by the high number of drugs or by the disease itself, forcing the patient to take all these medications [47]. Whereas few drugs seem to have a direct effect on the microbiota, i.e., metformin or proton pump inhibitors, the association of multiple compounds is not clearly associated to the depletion or enrichment of specific taxa. All the patients but one were taking allopurinol, a common urate-reducing drug. In a rodent model of hyperuricaemia, its use was associated with an increase in the relative abundance of *Bifidobacterium* spp. [48]. In our GSD patients, we did observe a slight increase in this genus, but this observation should be confirmed in a bigger cohort.

In conclusion, we believe that our study could pave the way for further investigations of the intestinal bacterial community in GSD type Ia and Ib patients and in similar metabolic syndromes. In the frame of glycogen storage diseases, studies evaluating gut microbiota differences in large multicentric cohorts are needed to expand our results obtained by a cohort limited by the rarity of the disease, albeit homogeneous. Nevertheless, our study showed how profoundly gut microbiota can be modulated by a life-long diet. Importantly, future studies should aim at clarifying whether the observed changes are driven by nutritional parameters only or also by the disease itself.

## 4. Materials and Methods 

### 4.1. Subject Recruitment and Sampling

For this study, 9 GSD type I patients (Ia = 4, Ib = 5) and 12 healthy controls (HC) were enrolled from January 2018 to June 2018. The dataset consisted of 21 subjects, gender and age matched between groups. Mean age of GSD patients was 27.7 ± 12.5 years (6 males and 3 females), while mean age of HC was 24.7 ± 7.9 years (9 males and 3 females). Despite only six subjects, three GSD and three HC, being of pediatric age, all the enrolled subjects in the study were followed by the Pediatric Department of San Paolo Hospital, reference center for metabolic diseases in Milan (Italy). For both patients and controls, inclusion criteria were: gestational age 37–42 week inclusive, birth weight ≥ 2500 g and single birth; exclusion criteria were: treatments with antibiotic and/or probiotic/prebiotic assumption during the previous 3 months.

Specific GSD inclusion criteria were: disease clinical onset during childhood and the diagnosis confirmed by liver biopsy (% hepatic glycogen and glucose-6-phosphatase enzymatic activity assay); dosage of deoxyglucose transport in polymorphonuclear neutrophils (only in patients with GSD Ib) and/or molecular analysis of GSD; to be on treatment with CS; not to have type I/II diabetes.

Stool samples, stored at −80 °C until use, were collected for each subject. Pediatricians performed anthropometric measurements (height, weight) and body mass index was calculated; the nutritional weight status was evaluated through the WHO classification of underweight, overweight and obese adult for patients >18 years [49], while for patients ≤ 18 years standard scores (z-scores) of BMI were calculated and evaluated using WHO reference standard [50].

Furthermore, a 24-h food recall was provided by patients themselves or by parents to collect dietary data. Dietary food records were processed by dieticians in order to calculate the average amounts of energy and nutrient intake (carbohydrates, soluble glucids, starch, soluble and insoluble fibers, lipids, proteins) using commercially available software (MetaDietaR, Software version 3.1, ME.TE.DA S.r.l., San Benedetto del Tronto, Italy). For each subject, the use of drugs was also evaluated.

In addition, in conjunction with the stool collection, we collected biochemical data of GSD patients from their routine check-up. The metabolic parameters evaluated were: glycemia, insulin (with HOMA-IR, HOMAβ, QUICKI and Tyg-Index calculation), total cholesterol, triglycerides, uric acid, lactate and transaminases. The HOMA-IR (homeostasis model assessment of insulin resistance) was calculated as follows: [basal blood glucose (mg/dl) × basal insulin (IU/mL)]/405; the QUICKI (quantitative insulin sensitivity index) was calculated as follows: 1/[log10 insulin (μUI/mL) + log10 basal blood glucose (mg/dL)]; HOMAβ (Homeostatic Model Assessment of β-cell function) was calculated as follows: 20 × basal insulin (μUI/mL)/[basal blood glucose (mmol/L) -3.5] [51]; the Tyg-index (triglycerides and glucose index) was calculated as follows: Log [triglycerides (mg/dL) × glycaemia (mg/dL)/2] [52].

The study was conducted at the Pediatric Department of San Paolo Hospital in Milan, with the previous approval by Ethics Committee of San Paolo Hospital in Milan (Comitato Etico Milano Area 1, Protocol number 2017/ST/13749); written informed consent was obtained from each enrolled subject.

### 4.2. Gut Microbial DNA Extraction and Sequencing

Fecal DNA extraction was performed using the Spin stool DNA kit (Stratec Molecular, Berlin, Germany), according to manufacturer’s instructions. For each sample, 25 ng of extracted DNA was used to construct the sequencing library.

The V3–V4 hypervariable regions of the bacterial 16S rRNA gene were amplified with a two-step barcoding approach according to the Illumina 16S Metagenomic Sequencing Library Preparation (Illumina, San Diego, CA, USA). Briefly, DNA samples were amplified with dual-index primers using a Nextera XT DNA Library Preparation Kit (Illumina) and library concentration and quantification were determined using a KAPA Library Quantification Kit (Kapa Biosystems, Woburn, MA, USA) and Agilent 2100 Bioanalyzer System (Agilent, Santa Clara, CA, USA), respectively. The libraries were pooled and sequenced with a MiSeq platform (Illumina) for 2 × 250 base paired-end reads and a total of 2.5 Gbases raw reads were obtained.

### 4.3. Microbiota Profiling

The obtained 16S rRNA gene paired sequences were merged using Pandaseq [53] (release 2.5). Reads were filtered by trimming stretches of 3 or more low-quality bases (quality < 3) and discarding the trimmed sequences whenever they were shorter than 75% of the original one.

Bioinformatic analyses were conducted using the QIIME [54] pipeline (release 1.8.0), clustering filtered reads into Operational Taxonomic Unit (OTUs) at 97% identity level and discarding singletons as possible chimeras. Taxonomic assignment was performed via the RDP classifier [55] against the Greengenes database [56] (v 13_8).

Alpha-diversity was computed using the chao1, number of OTUs, Shannon diversity and Faith’s phylogenetic diversity whole tree (PD whole tree) metrics through the QIIME pipeline; statistical evaluation among alpha-diversity indices was performed by a non-parametric Monte Carlo-based test, using 9999 random permutations. Weighted and unweighted UniFrac distances and Permanova (adonis function) in the R package vegan [57] (version 2.0–10) were used to compare the microbial community structure of GSD and HC subjects. A functional prediction of the bacterial metabolic pathways was performed using PICRUSt software [58] (version 1.0.1) and KEGG pathways database [59]. Differences in functional category profiles between breeds were assessed using Bray–Curtis distance among samples and “adonis” permutation-based test on the experimental labels.

### 4.4. Fecal Short Chain Fatty Acids Measurement

Fecal short chain fatty acids (SCFAs) quantification was performed by gas chromatography. Concentrations of acetic, propionic, iso-butyric, butyric and iso-valeric acids were assessed as previously described [60].

Briefly, analyses were performed using a Varian model 3400 CX Gas chromatograph fitted with FID detector, split/splitless injector and a SPB-1 capillary column (30 m × 0.32 mm ID, 0.25 μm film thickness; Supelco, Bellefonte, PA, USA). Quantification of the SCFAs was obtained through calibration curves of acetic, propionic, iso-butyric, butyric, and iso-valeric acid in concentrations between 0.25 and 10 mM (10 mM 2-ethylbutyric acid as internal standard). Results are expressed as mg/g of wet weight of feces.

### 4.5. Statistical Analysis

All comparisons were performed using MATLAB software (Natick, MA, USA; version 2008b).

Comparisons of the two groups were performed using Student’s t-test for normally distributed variables and Wilcoxon test for non-normally distributed variables. For evaluating differences in relative abundances of bacterial groups, a Mann–Whitney U-test was performed. Due to the small number of samples, no correction methods have been applied.

Correlations between taxa and nutritional values and SCFA quantities were assessed through Spearman correlation and the associated linear regression model. *p*-values < 0.05 were considered as significant for each analysis.

## Figures and Tables

**Figure 1 metabolites-10-00133-f001:**
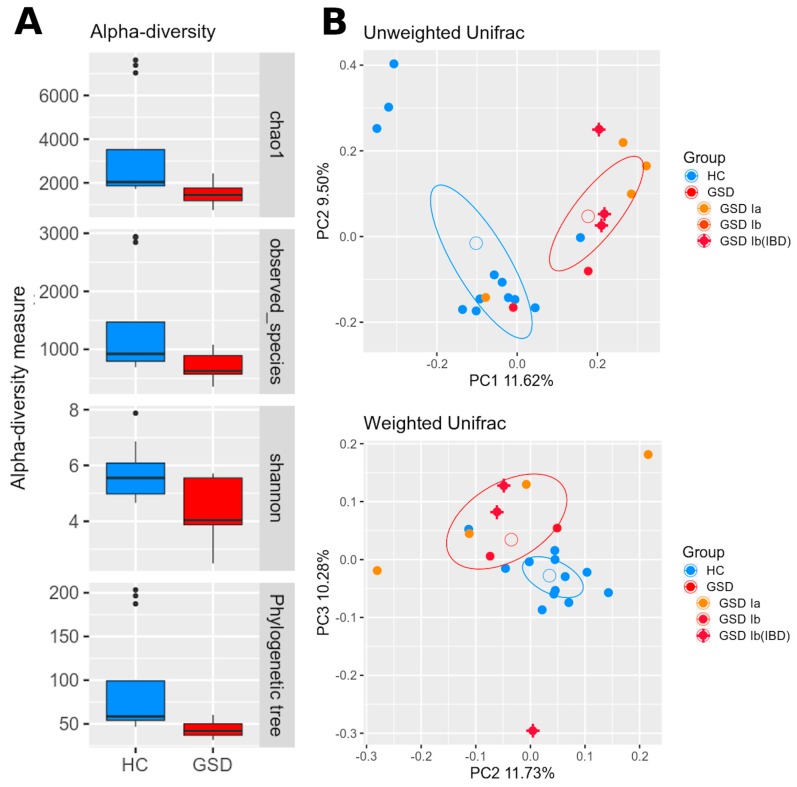
Biodiversity and phylogenetic analysis between cohorts. (**A**) Alpha-diversity indexes are reported for healthy control (HC) (blue) and glycogen storage disease (GSD) (red) subjects for chao1, observed species, Shannon diversity and Faith’s phylogenetic metrics. Diversity among groups is statistically significant for all metrics. (**B**) Beta-diversity analysis represented by PCoA graphs of weighted and unweighted UniFrac distance between HC (blue) and GSD (red) subjects. The ellipses of mean standard error (SEM)-based data confidence are reported. Microbial communities are statistically different for both distances (adonis test: unweighted *p* = 0.004; weighted *p* = 0.01). Percentage variance accounting for the first, second and third principal components is shown along the axis. To highlight possible differences related to GSD type, a color scheme was further applied to the GSD group: GSD-Ia (orange), GSD-Ib (red), GSD-Ib with inflammatory bowel disease (IBD) (red + cross).

**Figure 2 metabolites-10-00133-f002:**
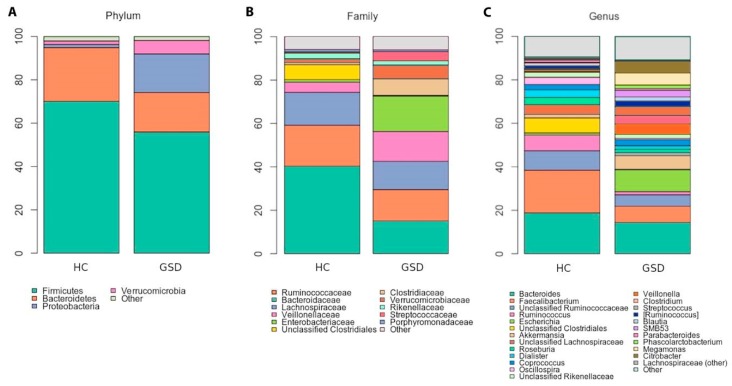
Taxonomic characterization. Stacked bar charts of taxonomy relative abundances at (**A**) phylum, (**B**) family and (**C**) genus levels for healthy controls (HC) and GSD patients. Only phyla, families and genera present at relative abundances >1% in at least 20% subjects (i.e.,: ≥ 4 samples) are reported. Remaining taxa are grouped in the “Other” category for each level.

**Figure 3 metabolites-10-00133-f003:**
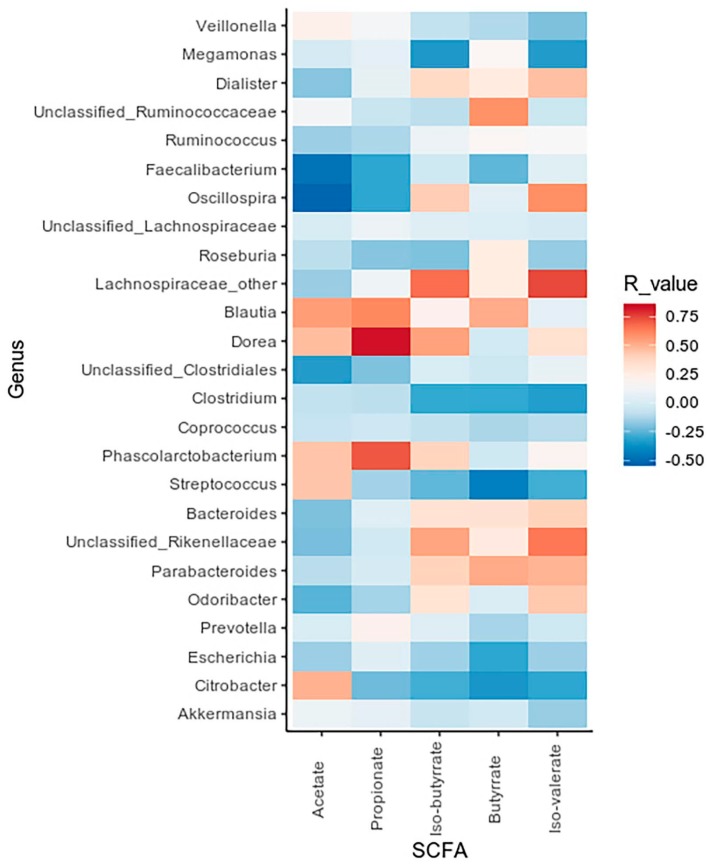
Correlation between SCFA values and bacterial genera. Heatmap showing Spearman’s correlations between the most abundant microbial genera and SCFA concentrations. Red tiles indicate a positive correlation, blue tiles a negative one for both HC and GSD groups.

**Figure 4 metabolites-10-00133-f004:**
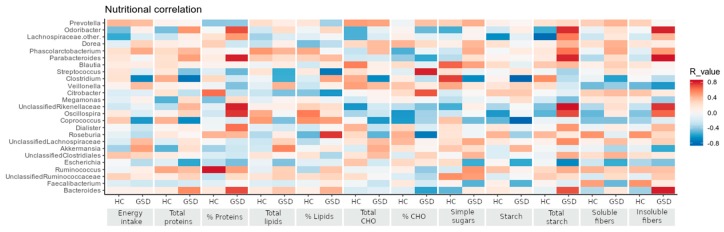
Correlations between nutritional and taxonomic values. Bacterial genera are reported in the same order of relative abundance as in Table 2, in correlation with the nutritional values shown in Table 1 according to Spearman’s correlation. Red tiles indicate a positive correlation, blue tiles a negative one for both HC and GSD groups.

**Table 1 metabolites-10-00133-t001:** Nutritional values of the two enrolled groups.

Nutritional Variable	HC	GSD	*p*-Value	
	Mean ± SD	Mean ± SD		
Energy Intake				
kcals	1907 ± 603	2420 ± 549	0.0468	*
Proteins				
g	74.67 ± 22.92	83.06 ± 20.56	0.3824	
% energy	16.70 ± 3.77	13.80 ± 2.20	0.0815	
Lipids				
g	77.96 ± 47.06	62.35 ± 15.47	0.7021	
% energy	36.58 ± 10.65	23.40 ± 3.26	0.0013	**
Carbohydrates				
g	216.19 ± 54.55	390.03 ± 97.78	0.0007	***
% energy	46.28 ± 9.01	60.22 ± 4.54	<0.0001	****
Sugars				
g	58.56 ± 25.44	23.75 ± 9.11	0.0013	**
% energy	11.98 ± 4.74	3.56 ± 1.06	<0.0001	****
Fiber				
overall, g	15.44 ± 4.80	21.01 ± 4.37	0.0148	*
overall, g/1000 kcal	8.58 ± 2.16	9.10 ± 2.72	0.7021	
insoluble fiber, g	6.43 ± 4.44	9.59 ± 4.43	0.1285	
soluble fiber, g	2.01 ± 1.35	3.57 ± 1.23	0.0227	*

Values are expressed as means (with standard deviations). Significant differences are indicated by * (*p*-value < 0.05), ** (*p*-value < 0.01), *** (*p*-value < 0.001) and **** (*p*-value < 0.0001), Mann–Whitney test.

**Table 2 metabolites-10-00133-t002:** Taxonomic relative abundance at the genus level.

Genus	Average Relative Abundance		*p*-Value
	HC	GSD	
*Bacteroides*	18.83	14.43	0.2410
*Faecalibacterium*	19.61	7.44	0.0209 *
*Unclassified Ruminococcaceae*	8.94	5.27	0.0700
*Ruminococcus*	7.25	1.42	0.0173 *
*Escherichia*	0.99	10.01	0.0077 **
*Unclassified Clostridiales*	6.87	0.29	0.0025 **
*Akkermansia*	1.63	6.26	0.2323
*Unclassified Lachnospiraceae*	4.48	1.41	0.0428 *
*Roseburia*	3.27	1.50	0.0428 *
*Dialister*	3.57	1.64	0.0360 *
*Coprococcus*	2.43	2.80	0.4138
*Oscillospira*	3.35	0.64	0.0020 **
*Unclassified Rikenellaceae*	2.35	1.87	0.1657
*Veillonella*	0.41	4.73	0.1265
*Clostridium*	0.76	3.94	0.4996
*Streptococcus*	0.51	4.16	0.1886
*Blautia*	1.40	1.83	0.4996
*SMB53*	0.29	3.05	0.0360 *
*Parabacteroides*	1.07	0.84	0.1658
*Phascolarctobacterium*	0.51	1.58	0.4946
*Megamonas*	0.00	5.64	0.0092 **
*Citrobacter*	0.07	5.33	0.3609
*Bifidobacterium*	0.23	0.70	0.7754
Other genera	10.68	13.41	--

The main genera in GSD patients and healthy controls, selected for <1% abundance in at least one of the two groups, are reported. All bacterial taxa present at less than 1% relative abundance were grouped into the “Other genera” classification. Significant differences are indicated by * (*p*-value < 0.05) and ** (*p*-value < 0.01).

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
