# Peer review of "Proteobacteria Overgrowth and Butyrate-Producing Taxa Depletion in the Gut Microbiota of Glycogen Storage Disease Type 1 Patients"

_metabolites, 2020, doi:10.3390/metabo10040133_

Round 1

Reviewer 1 Report

Ceccarani et al. describe in their manuscript the differences in the gut microbiome of patients with glycogen storage disease and healthy volunteers. The disease seems to be rare, and I have never heard of it before. Due to the impact this disease has on the availability of sugars, it is a prime candidate for gut microbiota investigation. I liked to read this paper. I have various smaller suggestions below.

  • a problem here is obviously that the effect of disease and medication cannot be distinguished. It would be good if this could be discussed, also in context with https://www.ncbi.nlm.nih.gov/pubmed/29555994?dopt=Abstract https://www.ncbi.nlm.nih.gov/pubmed/31953381
  • Table 1: I’d like to see p-values for everything based on absolute and relative amounts
  • The age of the patients needs to be better described. They were recruited at a pediatric unit, but some seem to have been 18+, which confuses me, and the age of the matched controls need to be described better too. Especially given that the microbiota changes multiple times over the life span which has been investigated https://www.ncbi.nlm.nih.gov/pubmed/22919693/
  • Line 133: Please read https://www.ncbi.nlm.nih.gov/pubmed/24699258 . The subsampling process is not described, if I see it right
  • Figure 2: The resolution seems insufficient (might be an artifact from the reviewing PDF), and a different colour scheme is necessary, because on C you cannot differentiate between the different yellows, beiges, etc.
  • also, for Fig2 B, are you sure you labelled the bars correctly? For me it seems you swapped GSD and HC
  • Figure 4 would maybe be better in the appendix (together with the data on which its based)
  • this paper https://www.ncbi.nlm.nih.gov/pubmed/28798125 should be highly relevant for the discussion. It seems as an increase in Enterobacteriaceae is a general sign of gut disturbance.
  • Line 253: Did anyone have a look if the IBD patients are actually different? Might be worth marking them in Figure 2
  • Line 257: So the GSD patients had higher starch consumption, but a lower amount of predicted starch degradation genes. Would this not maybe indicate that more sugar/fibre/etc is excreeted? The GSD subjects also consume more in total and have a higher BMI. Is maybe more directly absorbed by the host after bacterial breakdown?
  • Line 308: Might require a citation
  • line 367+: No controls used? Contamination is luckily not much of a problem for faecal material, but I’d strongly insist that you use positive and negative controls for further research https://www.ncbi.nlm.nih.gov/pubmed/30997495 , so that contaminants https://www.ncbi.nlm.nih.gov/pubmed/30497919 https://www.ncbi.nlm.nih.gov/pubmed/25387460 can be detected
  • Line 386: Version of the database
  • Line 387: Which tool was used for the metrics?
  • Line 406: I don’t see that multi-test correction is mentioned. While the results are described, a total overview is also necessary (how many ASVs/OTUs, how many stat significant)
  • Line 407: Version?
  • The raw sequencing data needs to be uploaded to the NCBI/EBI
  • It would be good if the biom file (as .tsv version) could be added as appendix
  • From supplementary table 1 it looks like both GSD groups are pretty different. There needs to be some discussion why they were further treated as one. You can also see in the PcoAs in Fig1 that the group is not homogenous, and I wonder what would happen if you put the subgroups in. For table 2 I would also like to see it split up in table 2.
  • Was by any chance lactate measured? Veillonella consumes lactate, so this could be interesting

I wish you also good luck with your research. I hope also that you are dealing well with the coronavirus and that your families are save.

Author Response

We would like to express our deep gratitude to both reviewers for helpful comments and positive feedback. According to reviewers' suggestions we have modified the manuscript and added new information (changes are highlighted in the manuscript) as explained in the point-by-point reply and we believe that the manuscript in this revised form is significantly improved.

Comment 1- a problem here is obviously that the effect of disease and medication cannot be distinguished. It would be good if this could be discussed, also in context with https://www.ncbi.nlm.nih.gov/pubmed/29555994?dopt=Abstract https://www.ncbi.nlm.nih.gov/pubmed/31953381

Answer 1- Thank you for raising this issue. We did comment on this aspect in the discussion section (lines 321-331).

C2- Table 1: I’d like to see p-values for everything based on absolute and relative amounts

A2- We amended Table 1 according to reviewer suggestions

C3- The age of the patients needs to be better described. They were recruited at a pediatric unit, but some seem to have been 18+, which confuses me, and the age of the matched controls need to be described better too. Especially given that the microbiota changes multiple times over the life span which has been investigated https://www.ncbi.nlm.nih.gov/pubmed/22919693/

A3- Despite most of the patients are not pediatric anymore, because of the early diagnosis in the pediatric age they are still followed by the Pediatric Department in San Paolo Hospital, specialized in inborn errors of metabolism. We have now better specified this part in the M&M section (lines 344-346)

C4-Line 133: Please read https://www.ncbi.nlm.nih.gov/pubmed/24699258 . The subsampling process is not described, if I see it right

A4- Thank you for the suggestion. We amended the sentence describing the subsampling process. We carefully read the suggested paper and we will consider the valuable information for all our next projects. 

C5- Figure 2: The resolution seems insufficient (might be an artifact from the reviewing PDF), and a different colour scheme is necessary, because on C you cannot differentiate between the different yellows, beiges, etc. also, for Fig2 B, are you sure you labelled the bars correctly? For me it seems you swapped GSD and HC

A5- We amended the Figures increasing the DPI. We hope that the embedding process within the text would not affect the resolution. Concerning figure 2, we thank the reviewer for the advice. Indeed, family histograms were inverted. We replaced the figure with the amended version (also improved with more colors).

C6- Figure 4 would maybe be better in the appendix (together with the data on which its based).

A6- We thank the reviewer for this comment. However, we already kept the figure numbers at a minimum and we feel that Figure 4 and the relative paragraph contain information worthy of the main text. 

C7- this paper https://www.ncbi.nlm.nih.gov/pubmed/28798125 should be highly relevant for the discussion. It seems as an increase in Enterobacteriaceae is a general sign of gut disturbance.

A7- We are grateful for this suggestion, as we have now expanded the discussion on this aspect (lines 277-284).

C8- Line 253: Did anyone have a look if the IBD patients are actually different? Might be worth marking them in Figure 2

A8- We highlighted in Figure 2 (by color code) samples from patients with GSD Ia, Ib, and Ib with IBD. Nevertheless, it does not seem that IBD is a clustering factor.

C9- Line 257: So the GSD patients had higher starch consumption, but a lower amount of predicted starch degradation genes. Would this not maybe indicate that more sugar/fibre/etc is excreted? The GSD subjects also consume more in total and have a higher BMI. Is maybe more directly absorbed by the host after bacterial breakdown? 

A9-  We thank the reviewer for this valuable suggestion. Unfortunately, we did not measure the excreted amount of sugars or fiber, but, due to the compromised efficiency of the microbiota of GSD patients, we speculate that the excreted amount of starch could be higher in GSD compared to controls. 

C10- Line 308: Might require a citation

A10- We moved citation 46, encompassing the role of Veillonellaceae in propionate production, to allow readers to have a clear reference.

C11- line 367+: No controls used? Contamination is luckily not much of a problem for faecal material, but I’d strongly insist that you use positive and negative controls for further research https://www.ncbi.nlm.nih.gov/pubmed/30997495 , so that contaminants https://www.ncbi.nlm.nih.gov/pubmed/30497919 https://www.ncbi.nlm.nih.gov/pubmed/25387460 can be detected

A11- We thank the reviewer for this important suggestion. Indeed, for samples other than stools, we always include negative controls to avoid misinterpretation of data from l a low microbial biomass samples (for example Borghi E, Massa V, Severgnini M, et al. Antenatal Microbial Colonization of Mammalian Gut. Reprod Sci. 2019;26(8):1045–1053).

C12- Line 386: Version of the database

A12- We added this information (line 398).

C13-Line 387: Which tool was used for the metrics?

A13-We added this information (line 400).

C14-Line 406: I don’t see that multi-test correction is mentioned. While the results are described, a total overview is also necessary (how many ASVs/OTUs, how many stat significant)

A14- P-values were not corrected. A sentence has been added to clarify this aspect (lines 422-423). The OTU counts have been added as suggested (line 132-135).

C15-Line 407: Version?

A15-We added this information (line 419).

C16- The raw sequencing data needs to be uploaded to the NCBI/EBI

It would be good if the biom file (as .tsv version) could be added as appendix

A16- We added in the “Supplementary Material section” the accession number for raw reads (lines 429-431).

C17-From supplementary table 1 it looks like both GSD groups are pretty different. There needs to be some discussion why they were further treated as one. You can also see in the PcoAs in Fig1 that the group is not homogenous, and I wonder what would happen if you put the subgroups in. 

A17- As suggested, we added a color code in Figure 1 for the different subgroups. Because of the highly interindividual variation and the small cohort, we believe it is better to discuss GSD Ia and Ib as a sole group.

C18- For table 2 I would also like to see it split up in table 2.

A18- As discussed above, because of the highly interindividual variation and the small cohort (driven from the rarity of the disease), splitting the table into 2 subgroups would not result in more informative data

C19- Was by any chance lactate measured? Veillonella consumes lactate, so this could be interesting

A19- Unfortunately, we did not measure fecal lactate as, due to the scarcity of some fecal samples, we decided to focus on the main SCFAs.

Reviewer 2 Report

General comments:

The article is generally well written, with very few grammatical errors and other formatting errors. Also, it is presented in a simplistic manner for a broad readership in this field. Overall, I will recommend the article to be published with the minor corrections that are presented in the Specific comments below. I hope that this article will have a good impact on many future studies in this field.

Specific comments:

Abstract: Line 19: replace the word 'assumption' by 'consumption'.

Introduction: Line 36: replace the term 'bacteria' by 'bacterial'

Results, Line 99: Remove the term '(SD)'. Replace '27.7 (12.5)' by '27.7 ± 12.5'

Line 100: same as line 99

Lines 98 - 109: This part should be in Methods and materials section

Line 135: what is the sequence identity? >97%? This should be indicated.

Line 142: Figure 1 and all other figures - Please make sure that the images included are at least 300 DPI. The current images look more like 72 DPI.

Line 253: The names of 'Escherichia coli', and others elsewhere in the manuscript should be italicized.

Author Response

We would like to express our deep gratitude to both reviewers for helpful comments and positive feedback. According to reviewers' suggestions we have modified the manuscript and added new information (changes are highlighted in the manuscript) as explained in the point-by-point reply and we believe that the manuscript in this revised form is significantly improved.

Comment 1- Abstract: Line 19: replace the word 'assumption' by 'consumption'. 

Answer 1- We amended the abstract.

C2-Introduction: Line 36: replace the term 'bacteria' by 'bacterial' 

A2- We rephrased the sentence (lines 36-39)

C3- Results, Line 99: Remove the term '(SD)'. Replace '27.7 (12.5)' by '27.7 ± 12.5' 

A3- We used throughout the text the “27.7 ± 12.5” format.

C4-Lines 98 - 109: This part should be in Methods and materials 

A4- We thank the reviewer for this suggestion: we have now moved this part in the M&M section (lines 341-344)

C5-Line 135: what is the sequence identity? >97%? This should be indicated.

A5- This information is reported at line 396: “Bioinformatic analyses were conducted using the QIIME pipeline (release 1.8.0), clustering filtered reads into Operational Taxonomic Unit (OTUs) at 97% identity level and discarding singletons as possible chimeras….

C6-Line 142: Figure 1 and all other figures - Please make sure that the images included are at least 300 DPI. The current images look more like 72 DPI.

A6- We amended the Figures increasing the DPI. We hope that the embedding process within the text would not affect the resolution.

C7-Line 253: The names of 'Escherichia coli', and others elsewhere in the manuscript should be italicized. 

A7- We strongly apologize for these typos. We have now italicized all the species, genus, and family names.